# Biological Therapy of Severe Asthma with Dupilumab, a Dual Receptor Antagonist of Interleukins 4 and 13

**DOI:** 10.3390/vaccines10060974

**Published:** 2022-06-19

**Authors:** Corrado Pelaia, Giulia Pelaia, Claudia Crimi, Angelantonio Maglio, Giuseppe Armentaro, Cecilia Calabrese, Angela Sciacqua, Luca Gallelli, Alessandro Vatrella

**Affiliations:** 1Department of Health Sciences, University “Magna Græcia” of Catanzaro, 88100 Catanzaro, Italy; giulia.pelaia@gmail.com (G.P.); gallelli@unicz.it (L.G.); 2Department of Clinical and Experimental Medicine, University of Catania, 95123 Catania, Italy; dott.claudiacrimi@gmail.com; 3Department of Medicine, Surgery and Dentistry, University of Salerno, 84084 Salerno, Italy; amaglio@unisa.it (A.M.); avatrella@unisa.it (A.V.); 4Department of Medical and Surgical Sciences, University “Magna Græcia” of Catanzaro, 88100 Catanzaro, Italy; peppearmentaro@libero.it (G.A.); sciacqua@unicz.it (A.S.); 5Department of Translational Medical Sciences, University of Campania “Luigi Vanvitelli”, 80131 Naples, Italy; cecilia.calabrese@unicampania.it

**Keywords:** severe asthma, IL-4, IL-13, dupilumab

## Abstract

Interleukin-4 (IL-4) and interleukin-13 (IL-13) are key cytokines involved in the pathophysiology of both immune-inflammatory and structural changes underlying type 2 asthma. IL-4 plays a pivotal role in Th2 cell polarization, immunoglobulin E (IgE) synthesis and eosinophil recruitment into the airways. IL-13 synergizes with IL-4 in inducing IgE production and also promotes nitric oxide (NO) synthesis, eosinophil chemotaxis, bronchial hyperresponsiveness and mucus secretion, as well as the proliferation of airway resident cells such as fibroblasts and smooth muscle cells. The biological effects of IL-4 and IL-13 are mediated by complex signaling mechanisms activated by receptor dimerization triggered by cytokine binding to the α-subunit of the IL-4 receptor (IL-4Rα). The fully human IgG4 monoclonal antibody dupilumab binds to IL-4Rα, thereby preventing its interactions with both IL-4 and IL-13. This mechanism of action makes it possible for dupilumab to effectively inhibit type 2 inflammation, thus significantly reducing the exacerbation of severe asthma, the consumption of oral corticosteroids (OCS) and the levels of fractional exhaled NO (FeNO). Dupilumab has been approved not only for the add-on therapy of severe asthma, but also for the biological treatment of atopic dermatitis and nasal polyposis.

## 1. Introduction

Asthma is a highly prevalent chronic airway disease, commonly featured by reversible bronchial obstruction associated with inflammation and remodeling of the respiratory tract [1,2,3]. The ‘umbrella’ term asthma includes many different phenotypes, arising from complex interactions between individual predisposing traits and environmental agents [4,5]. The various phenotypes are mostly driven by distinct airway inflammatory patterns, originating from pathobiologic intercellular connections named endotypes, which can also lead to the clinical expression of severe asthmatic variants [6,7]. In particular, the distinct phenotypes/endotypes of asthma can be characterized as either eosinophilic, neutrophilic, mixed, or paucigranulocytic profiles [8,9,10,11]. Eosinophilic airway inflammation is the most frequent pathophysiologic subtype of asthma, underpinned by type 2 (T2-high) allergic or non-allergic mechanisms and consisting of synergistic communications between innate and adaptive immune responses coordinated by both T helper 2 (Th2) lymphocytes and group 2 innate lymphoid cells (ILC2), which release interleukins 4 (IL-4), 13 (IL-13) and 5 (IL-5) [12,13,14]. In T2-high asthma, such cytokines are responsible for the development and amplification of airway inflammation and remodeling. In particular, IL-4 induces the maturation and expansion of the Th2 immunophenotype and also stimulates the production of immunoglobulins E (IgE), whilst IL-13 is mainly involved in the pathogenesis of mucus hypersecretion, bronchial hyperresponsiveness and airway remodeling (Figure 1) [15]. IL-5 is the key cytokine implicated in the differentiation, survival, and degranulation of eosinophils [16]. In this pathologic scenario, a pivotal deleterious action is exerted by environmental factors (e.g., aeroallergens, airborne pollutants, smoking, viral and bacterial infections) which damage the airway epithelial cells, thereby inducing them to secrete large quantities of alarmins [17]. The latter are innate cytokines including interleukin-25 (IL-25), interleukin-33 (IL-33) and especially thymic stromal lymphopoietin (TSLP), which behave as upstream inducers of innate and adaptive immune cellular responses underlying type 2 asthma [18]. Indeed, alarmins are engaged in direct activation of ILC 2, as well as in the effective stimulation of dendritic cell-mediated lymphocyte commitment toward the Th2 lineage [6,14]. As a result of such pathomechanisms, alarmins enhance the biosynthesis of type 2 cytokines, and among these IL-13 also promotes the release of IL-33, thus fostering a vicious circle which further expands type 2 asthma [19].

The above concepts make it possible to fully understand the efficacy of the biological treatments of type 2 severe asthma which target IgE, IL-5 or its receptor, IL-4 receptor and TSLP [20,21,22,23,24]. Within such a therapeutic context, the fully human monoclonal antibody dupilumab binds to the IL-4 receptor and suppresses the bioactivities of both IL-4 and IL-13; this mechanism explains why dupilumab can effectively dampen type 2 inflammation and provide significant clinical benefits in severe asthma (Figure 1), atopic dermatitis and nasal polyposis (Figure 1) [25,26,27].

On the basis of the previous arguments, the present narrative review will be focused on two main topics: (i) pathophysiologic functions of IL-4 and IL-13 in type 2 asthma; and (ii) the role of dupilumab as an add-on biological therapy of severe asthma.

## 2. Pathophysiologic Functions of IL-4 and IL-13 in Type 2 Asthma

The most relevant cellular elements which produce and release IL-4 and IL-13 are Th2 cells, T follicular helper (Tfh) lymphocytes and ILC2; other cell lines contributing to the biosynthesis of these two cytokines include mast cells, basophils, eosinophils, CD8^+^ T lymphocytes and natural killer cells [6,28,29,30,31,32]. With regard to type 2 allergic asthma, IL-4 and IL-13 are powerful inducers of airway inflammation and remodeling [6]. Initially secreted by basophils, IL-4 plays a key role in driving the commitment of naïve CD4^+^ T cells toward the Th2 lineage [33,34]. This essential pathobiologic function of IL-4 is also facilitated by its capacity to inhibit the immunomodulatory process mediated by regulatory T (Treg) lymphocytes [35], which physiologically suppress Th2 cell differentiation in normal subjects [36]. Acting together with IL-13, IL-4 represents the ‘first signal’ which stimulates IgE production by B lymphocytes [37,38]. Isotype switching leading to Ig rearrangement at the level of B lymphocytes also depends on a ‘second signal’ consisting of the so-called B-T cell cognate relationship expressed by an immunological synapse involving both antigen/T cell receptor (TCR) and CD40/CD40 ligand (CD40 L) interactions [39]. Though Th2 and Tfh lymphocytes are the main sources of IL-4 and IL-13, such cytokines can be also secreted by ILC2 [30]. The cross-linking resulting from allergen binding to adjacent IgE coupled with their high affinity receptors (FcεRI) elicits the degranulation of mast cells and basophils responsible for the release of copious quantities of IL-4 and IL-13 [6,40], which thus further expand type 2 airway inflammation.

Beyond B and T lymphocytes, within the airways IL-4 and IL-13 target other immune/inflammatory and resident cells implicated in asthma pathobiology. Indeed, IL-13 induces the proliferation of mast cells and increments their expression of FcεRI [41]. Activation of macrophage IL-4/IL-13 receptors leads to commitment towards the M2 macrophage lineage, which effectively participates in the pathophysiology of severe allergic asthma [42,43]. Moreover, IL-4 up-regulates the endothelial expression of vascular cell adhesion molecule-1 (VCAM-1), thus promoting eosinophil margination, whereas IL-13 stimulates eosinophil chemotaxis by eliciting the release of eotaxin-3 (CCL26) from bronchial epithelial cells [44,45]. The airway epithelial barrier is disrupted by IL-13 which suppresses the biosynthesis of claudin-18.1, an essential constituent of intercellular tight junctions [46]. Furthermore, IL-4 and IL-13 increase the expression of histone deacetylases 1 and 9 (HDAC 1 and 9), whose biological activities are inversely correlated to the integrity of bronchial epithelium [47]. Hence, IL-4 and IL-13 are powerful inducers of airway epithelial damage, which is a keynote of asthma pathogenesis. IL-13 drives the metaplasia of goblet cells and enhances the production of mucin 5AC (MUC5AC), a glycoprotein which confers high viscosity to bronchial mucus [48]. IL-13 also heightens the expression levels of the inducible isoform of NO synthase (iNOS) (Figure 1) thus raising FeNO [49], which is considered a relevant biomarker of type 2 airway inflammation. In addition, IL-13 promotes the contractile and proliferative responses of airway smooth muscle cells [22,29]. By augmenting fibroblast proliferation and collagen deposition, IL-13 also contributes to sub-epithelial fibrosis, which seems to be dependent at least in part on IL-13-mediated stimulation of transforming growth factor-β1 (TGF-β1) [50,51]. Taken together, all these actions exerted on airway structural cells by IL-13 explain its central role in the development of airway remodeling in asthma.

Large quantities of both IL-4 and IL-13 can be detected in blood, bronchial mucosa, induced sputum and bronchoalveolar lavage fluid (BALF) obtained from asthmatic patients [29,52,53]. Furthermore, allergen challenge up-regulates the airway mRNA levels of these cytokines [29,52,53,54]. Genetic studies have also identified interesting associations between IL-13/IL-13 receptor gene polymorphisms and bronchial hyperresponsiveness [55]. In particular, the individual susceptibility to develop asthma could be linked to specific polymorphisms detected in the RAD50-IL-13 region of chromosome 5q31.1 [56]. Despite some similarities between the bioactivities of IL-4 and IL-13, however, each of these cytokines also plays distinct pathogenic roles in asthma. IL-4 is determinant for Th2 cell polarization, whereas IL-13 is a pivotal inducer of mucus hypersecretion and airway remodeling [38,57,58]. When compared to IL-4, greater amounts of IL-13 can be observed in endobronchial biopsies from asthmatic patients, as well as in allergen-challenged murine lungs [59,60].

The biological actions of IL-4 and IL-13 are dependent on the activation of two receptor types located on many target cells [61,62,63]. IL-4 specifically interacts with the α-subunit of its receptor (IL-4Rα), which recruits the γC chain, thus leading to the assembly of the dimeric type I receptor (IL-4Rα/γC) [61]. IL-13 binds to the α1 subunit of the IL-13 receptor (IL-13Rα1), which associates with IL-4Rα to form the dimeric type II receptor (IL-4Rα/IL-13Rα1) [64]. The interaction between IL-4 and the type I receptor induces the recruitment of Janus kinases 1 (JAK 1) and 3 (JAK 3), which phosphorylate the tyrosine residues Y500, Y575, Y603 and Y633 located within the intracellular portion of the IL-4Rα subunit that in turn engage important signaling molecules such as insulin receptor substrate-2 (IRS-2) and signal transducer and activator of transcription-6 (STAT-6) [65,66,67,68,69]. These events lead to the activation of the phosphoinositide-3 kinase (PI3K) pathway implicated in the proliferative response of Th2 lymphocytes [25,68,69]. Moreover, JAK1/JAK3-dependent tyrosine phosphorylation promotes the dimerization and migration to the nucleus of STAT-6, which via the activation of the transcription factor GATA-3 enhances the expression of the genes coding for IL-4, IL-5, and IL-13 [25,70,71,72]. The type II receptor includes the IL-4Rα and IL-13Rα1 subunits and upon stimulation by both IL-4 and IL-13 activates JAK1/2, tyrosine kinase 2 (Tyk2) and STAT-6, [73,74]. IL-13 can also interact with the IL-13 receptor α2-chain (IL-13Rα2), which behaves as an inhibitory decoy receptor [75].

The main biological effects exerted by IL-4 and IL-13 within the asthmatic airways are summarized in Table 1. 

## 3. Role of Dupilumab as Add-On Biological Therapy of Severe Asthma

The fully human IgG4 monoclonal antibody dupilumab was developed by Regeneron Pharmaceuticals (Tarrytown, NY, USA) and Sanofi (Gentilly, France) [76,77]. Dupilumab is a dual receptor antagonist which occupies and blocks IL-4Rα, thus preventing its dimerization with the other components of type I and type II receptors, whose assembly is essential for the biological activities of IL-4 and IL-13 (Figure 1) [68,78]. 

Several phase 1 trials were carried out in healthy people with the aim of evaluating the safety, tolerability, immunogenicity and pharmacokinetics of dupilumab [77]. This biologic drug is characterized by a non-linear pharmacokinetic profile consisting of higher than dose-proportional increases in systemic exposure [77,79]. After being administered subcutaneously at a dosage of 600 mg, dupilumab exhibited a bioavailability of 64%, a distribution volume of 4.8 L and an average peak concentration of 70.1 μg/mL after one week [78].

An initial phase 2 a double-blind, randomized, and placebo-controlled trial was performed to verify the effects of dupilumab in 104 patients distributed across an age spectrum ranging from 18 to 65 years, complaining of moderate-to-severe asthma and displaying blood eosinophils counts of at least 300 cells/μL, as well as sputum eosinophil percentages of at least 3% [80]. The inhaled therapy included medium to high dosages of corticosteroids (ICS) and long-acting β_2_-adrenergic agonists (LABA), which were not able to guarantee an adequate asthma control. A total of 52 subjects received a placebo, and the remaining 52 patients underwent a biological therapy with 300 mg of dupilumab administered every week for 12 weeks. Treatment with LABA was stopped after 4 weeks, whilst ICS therapy was gradually reduced and then discontinued between the fourth and the ninth week. The main endpoint of this trial was to assess the changes in asthma exacerbations induced by dupilumab. A total of 23 individuals (44%) in the placebo arm and 3 patients (6%) assigned to dupilumab therapy reported an asthma exacerbation, respectively; this difference was shown to be statistically significant (*p* < 0.001), and dupilumab decreased the disease exacerbation rate by 87%. With regard to the secondary outcomes, dupilumab elicited a higher than 200 mL increase in forced expiratory volume in one second (FEV_1_) and also enhanced the morning peak expiratory flow (PEF). Moreover, dupilumab improved the score of the asthma control questionnaire (ACQ), decreased nocturnal awakenings, as well as morning and evening symptoms, and also reduced the use of as-needed short-acting bronchodilators. Additionally, dupilumab significantly decreased the levels of various biomarkers of type-2 asthma such as IgE, fractional exhaled nitric oxide (FeNO), eotaxin-3 (CCL26) and TARC (thymus and activation-regulated chemokine). In contrast, dupilumab augmented blood eosinophil counts in four subjects. When compared to a placebo, dupilumab increased the occurrence of injection-site reactions, nasopharyngitis, nausea, and headache. Only one patient reported the appearance of a cutaneous rash, which rapidly resolved after a bland therapy with antihistamines and systemic corticosteroids. Despite the useful information provided by this study, its experimental protocol was not consistent with treatment schedules implemented in daily clinical practice [78,81]. Indeed, in real-life management of asthma add-ons, biological therapies with monoclonal antibodies are always associated with a continuous ICS/LABA treatment.

Such a discordance was abolished by a subsequent larger trial consisting of a randomized, double-blind, placebo-controlled, dose-ranging, and parallel-group phase 2 b design, conducted in adult asthmatics not satisfactorily controlled by medium-to-high dosages of ICS/LABA combinations, which were not suspended during dupilumab treatment [82]. This trial consisted of three phases, including a preliminary screening lasting from 14 to 21 days, followed by a subsequent randomized 24-week treatment and by a final 16-week post-therapy monitoring period. A total of 776 people were randomly assigned to five subgroups treated as follows: (i) placebo for 158 patients; (ii) 200 mg of dupilumab every 2 weeks for 150 patients; (iii) 200 mg of dupilumab every 4 weeks for 154 patients; (iv) 300 mg of dupilumab every 2 weeks for 157 patients; (v) 300 mg of dupilumab every 4 weeks for 157 patients. With the exception of the third subgroup, when compared to the placebo arm all other patients treated with dupilumab enjoyed significant FEV_1_ increases, which after 24 weeks ranged from 0.15 to 0.16 L. Furthermore, when administered every 2 weeks, dupilumab significantly lowered the annual rate of severe asthma exacerbations. All dupilumab dosage schedules significantly reduced FeNO levels, which reached the lowest values when dupilumab was injected at intervals of 2 weeks. Dupilumab exerted its effects on asthma exacerbations, lung function and FeNO levels regardless of blood eosinophil numbers. This biologic drug also exhibited a good pattern of safety and tolerability, documented by similar incidences of mild adverse reactions throughout the five subgroups. Transient increments of blood eosinophil counts were found in some subjects who had baseline levels of at least 300 cells/μL.

The phase 3 LIBERTY ASTHMA QUEST study, a double-blind, placebo-controlled, randomized, and parallel-group trial, enrolled 1902 patients (at least 12 years old) with the aim of evaluating the efficacy of dupilumab in moderate-to-severe uncontrolled asthma [83]. The recruited subjects were randomly subdivided to receive one of the following subcutaneous treatments, lasting for 52 weeks: (i) initial loading dose of dupilumab 400 mg, followed by 200 mg every 2 weeks; (ii) matched volume of placebo, amounting to 1.14 mL; (iii) initial loading dose of dupilumab 600 mg, followed by 300 mg every 2 weeks; (iv) matched volume of placebo, amounting to 2 mL. When compared to a placebo, patients treated with dupilumab reported a nearly 50% reduction in the annualized rate of severe asthma exacerbations. Such a decrease in percentage was higher than 65% when the blood eosinophil number was at least 300 cells/μL. At both dosage schedules, dupilumab significantly increased FEV_1_, which reached its highest increments in patients with blood eosinophils counts of at least 300 cells/μL. Dupilumab also ameliorated asthma symptom control, as demonstrated by the relevant decreases of the scores related to the asthma control questionnaire (ACQ). Moreover, dupilumab significantly reduced many biomarkers of type 2 asthma, including FeNO, serum IgE concentrations and also the blood levels of periostin, eotaxin-3 and TARC. Temporary increases of blood eosinophil numbers were found in 52 patients (4.1%) treated with dupilumab, and in 4 patients (0.6%) assigned to the placebo arm. A post hoc evaluation of this study demonstrated that dupilumab-induced improvements in severe asthma exacerbations and type 2 asthma biomarker levels, as well as in asthma symptom control and pulmonary function, occurred independently of the atopic status [84].

The primary endpoint of the phase 3, double-blind, randomized, and placebo-controlled LIBERTY ASTHMA VENTURE trial was to verify the potential efficacy of dupilumab in reducing the daily intake of oral corticosteroids (OCS) [85]. A total of 210 participants with OCS-dependent severe asthma were recruited and randomized to receive for 24 weeks, according to a 1:1 ratio, either a placebo or dupilumab (initial dose of 600 mg, followed by 300 mg every two weeks). Dupilumab induced a 70.1% reduction in OCS consumption, which resulted to be significantly greater than the 41.9% decrease observed in the placebo arm. OCS therapy was suspended by 48% of subjects treated with dupilumab, and by 25% of patients who received a placebo. Despite the progressive reduction or even the complete interruption of OCS utilization, when compared to the placebo, dupilumab decreased by 59% the rate of severe asthma exacerbations and also elicited a 220 mL FEV_1_ increase. A transient elevation of blood eosinophil numbers was found in 14% of people undergoing treatment with dupilumab and in 1% of patients assigned to the placebo arm.

Many patients shifted from the above studies towards the TRAVERSE open-label extension trial, which enrolled 2282 adults and adolescents with moderate-to-severe asthma, recruited in 362 hospitals across 27 countries, and treated for 96 weeks with an initial loading dose of dupilumab (600 mg), followed by a dose of 300 mg every 2 weeks [86]. The results of this study further validated the therapeutic benefits provided by dupilumab in severe asthma, including important improvements in respiratory symptoms, disease exacerbations, and lung function. The main endpoint of the TRAVERSE trial was to assess the long-term safety profile of dupilumab, which appeared to be quite good. The most frequent side effects included nasopharyngitis, bronchitis and skin reactions at the injection site. The most common serious adverse events, affecting a very low number of patients, were pneumonia and asthma exacerbations. The anti-drug antibodies (ADAs) which were found in 157 participants did not affect the safety and efficacy of dupilumab. When considering the biomarkers of type 2 asthma, dupilumab gradually reduced the levels of both serum total IgE and blood eosinophils. 

The phase 3 double-blind, randomized, and placebo-controlled LIBERTY ASTHMA VOYAGE study assessed for 52 weeks the clinical and functional effects of dupilumab in children with moderate-to-severe asthma, aged 6 to 11 years [87]. A total of 408 participants received dupilumab or a placebo. Dupilumab was injected every 2 weeks at a dosage of 100 mg to patients with a body weight ≤ 30 kg and at a dosage of 200 mg to patients with a body weight > 30 kg, respectively. In comparison to the placebo, dupilumab significantly lowered the annualized rate of severe asthma exacerbations, as well as improved the predicted pre-bronchodilator FEV_1_ (ppFEV_1_) and the score of Asthma Control Questionnaire 7 Interviewer-Administered (ACQ-7-IA). Moreover, the toxicity profiles displayed by dupilumab and the placebo were quite similar.

The therapeutic properties of dupilumab have also been recently evaluated in real-life investigations. In this regard, a real-world multi-centre retrospective study, carried out across France in 64 patients with severe asthma, showed that after 12 months of treatment, dupilumab decreased the annual number of asthma exacerbations from four to one and increased the score of the asthma control test (ACT) from 14 to 22 [88]. Dupilumab also lowered OCS daily dosage from 20 to 5 mg and enhanced the median value of FEV_1_ from 58% to 68% of predicted value. Cutaneous reactions at the injection site were the most common unwanted effects. Similar results were obtained during a 1-year real-life multicentre study conducted in Italian patients with severe asthma [89].

Within a real-life context, we assessed in patients with severe asthma and nasal polyposis the onset of dupilumab effects, which were evaluated at the fourth week of treatment [90]. Dupilumab induced a rapid amelioration of subjective symptoms caused by both severe asthma and nasal polyposis, documented by significant improvements of ACT score (mean values: from 12 to 21) and sino-nasal outcome test-22 (SNOT-22) score (mean values: from 58 to 19). These positive clinical results made it possible to gradually lower and finally abolish OCS intake over a short period of 4 weeks. In addition, during the same time span, we recorded relevant increments of FEV_1_ (more than 200 mL), peak expiratory flow (0.6 L/s) and mid-expiratory flow at 25–75% of forced vital capacity (0.3). Such important functional changes were associated with marked decreases of pulmonary hyperinflation as shown by significant reductions of residual volume (−690 mL) and total lung capacity (−460 mL). These effects of dupilumab are really noticeable given the high relevance of air trapping in severe asthma [91]. 

Dupilumab also exerts effective therapeutic actions in asthma comorbidities including atopic dermatitis and nasal polyposis, underpinned by type-2 inflammation [26]. In this regard, the two phase 3 trials named LIBERTY NP SINUS-24 and -52 proved that dupilumab bettered nasal obstruction and congestion, as well as reduced the size of nasal polyps and the opacification of paranasal sinuses [92]. Furthermore, patients with both asthma and atopic dermatitis responded to dupilumab therapy with improvements in lung function and reductions of the eczema area [93]. Upon demonstration of dupilumab efficacy also in regard to the biological therapy of eosinophilic esophagitis [94], this monoclonal antibody has been recently (21 May 2022) approved in the United States (US) by the Food and Drug Administration (FDA) for such a new indication.

The main studies aimed to evaluate the effects of dupilumab in the biologic treatment of severe asthma are summarized in Table 2. The current indications for the therapeutic uses of dupilumab are summarized in Table 3.

## 4. Concluding Remarks

The recent progress in our understanding of the phenotypes and endotypes of severe asthma is offering great opportunities for the add-on biological therapies with monoclonal antibodies. In particular, by blocking at the receptor level the pathogenic actions of both IL-4 and IL-13 within the context of type 2 inflammation, dupilumab behaves as a very effective biologic drug for the treatment of severe asthma. In particular, specific indications for prescription of dupilumab include blood eosinophils counts of at least 150 cells/μL and/or FeNO levels of at least 25 ppb, especially when associated with OCS-dependence, nasal polyposis or atopic dermatitis [95]. Dupilumab can now be used in the US also for the biological treatment of eosinophilic esophagitis. 

## Figures and Tables

**Figure 1 vaccines-10-00974-f001:**
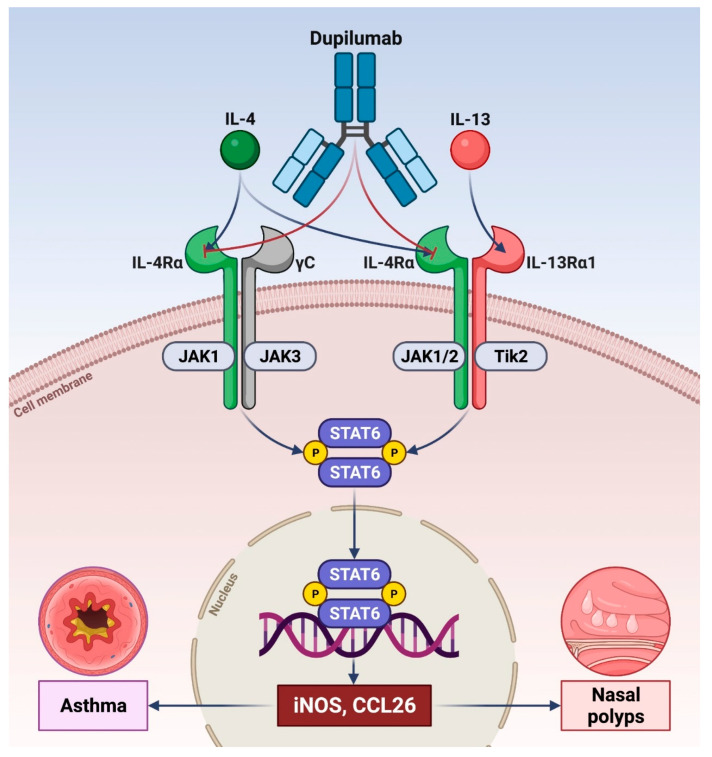
IL-4/IL-13 dual receptor blockade by dupilumab. Dupilumab is a fully human monoclonal antibody which binds to the α subunit of the IL-4 receptor, thus blocking at the receptor level the biological effects of IL-4 and IL-13, which activate a JAK/STAT-mediated signaling network involved in the pathogenesis of type 2 airway inflammation underlying asthma and nasal polyposis. IL-4Rα: α subunit of IL-4 receptor; IL-13Rα1: α1 subunit of IL-13 receptor; JAK: Janus kinase; STAT: signal transducer and activator of transcription; iNOS: inducible form of nitric oxide synthase; CCL 26: eotaxin−3. This original figure was created by the authors using “BioRender.com”.

**Table 1 vaccines-10-00974-t001:** Main biological actions exerted by IL-4 and IL-13 within the asthmatic airways.

Cytokine	Cellular Target	Main Effects
IL-4	Th cells	Commitment to Th2 cell lineage
IL-4 and IL-13	B cells	IgE isotype switch
IL-4 and IL-13	Eosinophils	Cell trafficking towards inflammatory sites
IL-4 and IL-13	Airway epithelial cells	Disruption of epithelial layer, up-regulation of iNOS
IL-13	Goblet cells	Mucus production
IL-13	Airway fibroblasts	Cell proliferation
IL-13	Airway smooth muscle cells	Increased contractility and proliferation

**Table 2 vaccines-10-00974-t002:** Dupilumab: summary of the main studies carried out in asthmatic patients.

Trial Name	Duration	Main Results or Endpoints
LIBERTY ASTHMA QUEST [83]	52 weeks	Fewer asthma exacerbations, better ACQ score, FEV_1_ increase, lower FeNO levels.
LIBERTY ASTHMA VENTURE [85]	24 weeks	Decreased OCS intake, fewer asthma exacerbations, FEV_1_ increase.
TRAVERSE [86]	96 weeks	Long-term safety and efficacy.
LIBERTY ASTHMA VOYAGE [87]	52 weeks	In children: fewer asthma exacerbations, better ACQ score, FEV_1_ increase.

**Table 3 vaccines-10-00974-t003:** Approved indications for the therapeutic uses of dupilumab.

Disease	Year of Approval	References
Severe asthma	2021	[80,81,82,83,84,85,86,87]
Nasal polyposis	2019	[92]
Atopic dermatitis	2020	[93]
Eosinophilic esophagitis	2022	[94]

## Data Availability

Not applicable.

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
