# Peer review of "Biological Therapy of Severe Asthma with Dupilumab, a Dual Receptor Antagonist of Interleukins 4 and 13"

_vaccines, 2022, doi:10.3390/vaccines10060974_

Round 1

Reviewer 1 Report

In this review the authors have tried to summarize important parts of type-2 inflammation, especially those relevant for asthma, as well as the results achieved so far with the biological agent dupilumab. The review is well-written, comprehensive and effective in its style. This reviewer has some suggestions for improvement.

COMMENTS

1. As mentioned, the review is very effective in its style but the problem is that primarily other review articles are cited. It would be good to include more original articles, at least pivotal work.

2. I have a problem with Figure 1. I do not agree that the main transcriptional effect of STAT6 is to upregulate the formation of the type-2 cytokines. These are mainly regulated by GATA-3, which can be activated by, for example T cell receptor activation and IL-33. Instead, the activation of STAT6 leads to increased expression of iNOS and CCL26, for example, besides inducing other effects in epithelial cells.

3. Introduction, line 41: the expression “diffuse” in this context is difficult to understand.

4. Section 2, lines 100-101: this does not seem to be included in Figure 1?

Author Response

1. As mentioned, the review is very effective in its style but the problem is that primarily other review articles are cited. It would be good to include more original articles, at least pivotal work.

With regard to citations, in the revised manuscript some review articles have been replaced as references by seminal original papers (reference numbers: 34,37,38,45,63,68).

2. I have a problem with Figure 1. I do not agree that the main transcriptional effect of STAT6 is to upregulate the formation of the type-2 cytokines. These are mainly regulated by GATA-3, which can be activated by, for example T cell receptor activation and IL-33. Instead, the activation of STAT6 leads to increased expression of iNOS and CCL26, for example, besides inducing other effects in epithelial cells.

In the revised manuscript, Figure 1 has been modified according to this suggestion.

3. Introduction, line 41: the expression “diffuse” in this context is difficult to understand.

In the revised text, the word “diffuse” has been replaced by “frequent”.

4. Section 2, lines 100-101: this does not seem to be included in Figure 1?

This referral to Figure 1 has been deleted in the revised manuscript.

Reviewer 2 Report

The authors present an excellent revision about the role of dupilumab as biological therapy of severe asthma. In particular, dupilumab behaves by blocking IL-4 and IL-13 receptors inhibiting type 2 inflammation and reducing significantly the exacerbations of severe asthma. Furthermore, dupilumab bettered nasal obstruction, lung function and reduced nasal polyps.

I have no suggestions for the authors because is an excellent review.

I read the review presented by Pelaia et. al. thoroughly and with great interest since the topic they present is closely related to my area of work.

As I was reading I could appreciate how well written the paper is, both from a formal and scientific point of view. Pelaia et al. present a review of the usefulness and results of phase 1 to phase 3 trials of the dupilumab antibody in the treatment of allergic asthma.

Several of the results presented are derived from the work done by the authors themselves, already published and reviewed as shown in references 26, 27 and 91.

The rest of the results presented are supported by the publications of other authors, and Pelaia et al. have been able to clearly and concisely reflect all these advances.

That is why I really have no objections from a scientific point of view to this excellent work which demonstrates that the authors are experts in the subject presented.

Author Response

We would like to thank very much this reviewer for his/her kind appreciation of our manuscript.

Reviewer 3 Report

1.  It would be helpful to add a table summarizing the physiologic functions of IL-4 and IL-13 that are discussed in section 2.

2. It would be helpful to add a table summarizing the role of dupilumab as add-on biologic therapy in severe asthma (discussed in section 3).

3. The next to last paragraph of section 3 would also benefit from the addition of a table summarizing these findings. 

4.  Dupilumab is now FDA approved for eosinophilic esophagitis in the US.  The last paragraph of section 3 and the paragraph in section 4 should be revised to reflect this additional indication. 

Author Response

1.  It would be helpful to add a table summarizing the physiologic functions of IL-4 and IL-13 that are discussed in section 2.

This table has been added in the revised manuscript.

2. It would be helpful to add a table summarizing the role of dupilumab as add-on biologic therapy in severe asthma (discussed in section 3).

This table has been added in the revised manuscript.

3. The next to last paragraph of section 3 would also benefit from the addition of a table summarizing these findings.

This table has been added in the revised manuscript.

4.  Dupilumab is now FDA approved for eosinophilic esophagitis in the US.  The last paragraph of section 3 and the paragraph in section 4 should be revised to reflect this additional indication.

In the revised manuscript these paragraphs have been changed according to such suggestions. 

Round 2

Reviewer 1 Report

I believe the appropriate amendments have been made to the manuscript.